# School-Based Circuit Training Intervention Improves Local Muscular Endurance in Primary School Students: A Randomized Controlled Trial

**DOI:** 10.3390/children10040726

**Published:** 2023-04-14

**Authors:** Nikola Stojanović, Dušan Stupar, Miroslav Marković, Nebojša Trajković, Dragana Aleksić, Goran Pašić, Admira Koničanin, Marko Zadražnik, Toplica Stojanović

**Affiliations:** 1Faculty of Sport and Physical Education, University of Niš, 18000 Niš, Serbia; nikola987_nish@hotmail.com (N.S.); nele_trajce@yahoo.com (N.T.); 2Faculty of Sports and Psychology, Educons University, 21000 Novi Sad, Serbia; 3Faculty of Sport, University “Union—Nikola Tesla”, 11000 Belgrade, Serbia; miroslav.markovic@fzs.edu.rs; 4Faculty of Sport and Physical Education, University of Priština—Kosovska Mitrovica, 38218 Leposavić, Serbia; dragana.aleksic@pr.ac.rs (D.A.); or toplica.stojanovic@ffvs.unibl.org (T.S.); 5Faculty of Physical Education and Sport, University of Banja Luka, 78101 Banja Luka, Bosnia and Herzegovina; goran.pasic@ffvs.unibl.org; 6Department of Biomedical Science, Sports, and Physical Education, State University of Novi Pazar, 36300 Novi Pazar, Serbia; akonicanin@np.ac.rs; 7Faculty of Sport, University of Ljubljana, 1000 Ljubljana, Slovenia; marko.zadraznik@fsp.uni-lj.si

**Keywords:** circuit training, local muscular endurance, primary school students

## Abstract

Background: This study aimed to explore the effectiveness of a 12-week circuit training program in improving local muscular endurance in normal-weighted primary school students. Methods: The study involved a parallel-group randomized trial with 606 primary school boys assigned to an experimental or a control group. The participants underwent a 12-week circuit training program that involved multi-joint, total-body workouts using body weight, resistance bands, and medicine ball exercises. The study measured the participants’ local muscular endurance during sit-ups, dynamic trunk extensions on a Roman bench (DTE), and push-ups. Results: After adjusting for the baseline, the treatment–grade interaction was significant for sit-ups (*F* = 7.74, *p* < 0.001, *η_p_*^2^ = 0.04), DTE (*F* = 6.49, *p* < 0.001, *η_p_*^2^ = 0.03), and push-ups (*F* = 9.22, *p* < 0.001, *η_p_*^2^ = 0.05), where the experimental treatment was more beneficial than the control. The treatment effect seemed to vary depending on the individual’s baseline local muscle endurance capacity. As the baseline local muscular endurance values increased, the treatment and grade effects became less beneficial. Conclusion: A 12-week circuit training program involving body weight, resistance bands, and medicine ball exercises suits school-based programs and can improve local muscular endurance in normal-weighted primary school boys. The experimental treatment was more effective than the control, and the individual baseline muscular endurance should be considered when designing training programs.

## 1. Introduction

Physical education is a crucial part of every school’s curriculum, as it has the potential to provide students with physical fitness, health, and social interaction skills [1]. However, there is uncertainty surrounding the amount of physical activity required to achieve these benefits and whether modern physical education can fulfill these requirements. Research suggests that school-based physical activity should be extended to at least 60 min per day [2,3], which exceeds the current standard of 3 45 min physical education classes in public schools in Serbia. Moreover, limited time, a lack of resources, unsupportive settings, and conflicting priorities are common obstacles that physical education teachers face in delivering effective and engaging classes [4]. These organizational issues may reduce students’ physical activity levels, adversely affecting their muscular fitness. Thus, exploring strategies to address these challenges and promoting physical activity among students during physical education classes is essential.

In addition to promoting overall physical activity, it is also essential for physical education programs to focus on developing specific aspects of physical fitness. Smith et al. [5] argue that cardiorespiratory fitness, flexibility, strength, local muscular endurance, and body composition are all key factors that strongly influence health outcomes. Specifically, building muscular endurance can result in a more resilient musculoskeletal system, reducing sports-related injuries, and improving overall health-related fitness [5]. However, despite the existence of comprehensive school-based programs in Serbia designed to improve various aspects of physical fitness [6], there is a concern that the current physical education curriculum may not place enough emphasis on abdominal musculature development [7,8]. Therefore, incorporating exercises involving rotational and diagonal patterns, using simple and inexpensive equipment like resistance bands and medicine balls, could be an effective strategy to enhance physical education programs in Serbia and to promote better muscular endurance in students.

A possible solution is to introduce a circuit training program targeting muscular endurance within physical education classes that, considering the limited resources, could increase students’ physical activity and improve overall local muscular endurance [9,10,11,12]. Faigenbaum, Kraemer, Blimkie, Jeffreys, Micheli, Nitka, and Rowland [9] suggest that bodyweight circuit training can be beneficial even if the equipment is unavailable, and in addition to enhancing muscular strength, power, and endurance, frequent involvement in a youth resistance training program can alter various other fitness-related variables [13]. Likewise, resistance training may positively affect some anatomical and psychological characteristics, prevent injuries in sports and leisure activities, and enhance motor abilities and athletic performance. Therefore, circuit training emphasizing muscular endurance seems advantageous as it provides various benefits.

It is advised that the focus is placed on working sets with more than 12 repetitions when developing programs to increase local muscle endurance [14]. When recommending an appropriate training intensity for children and adolescents, setting a repetition range between 10 and 15 and controlling the training load to maintain the intended intensity seems reasonable [8]. However, it is essential to note that resistance training should not be the sole focus of a physical education program. Faigenbaum, Kraemer, Blimkie, Jeffreys, Micheli, Nitka, and Rowland [9] recommend integrating team sports, swimming, distance running, cycling, and strength training in children’s and adolescents’ training programs. Therefore, local muscular endurance training programs should be time efficient to provide enough time for other physical education curriculum contents.

Strength training interventions for children and adolescents have been the subject of several studies to determine their effectiveness in improving muscular fitness and perceived physical competence. Faigenbaum et al. [15] compared the effects of low-repetition heavy-load and high-repetition moderate-load resistance training on children’s muscular strength and endurance development. The study showed that both interventions improved leg extension strength and endurance compared to the control group, but the high-repetition moderate-load training program was more effective in improving leg extension muscular endurance and chest press strength and endurance. Recently, Villa-González et al. [16] conducted a systematic review and meta-analysis to examine the effectiveness of school-based exercise interventions for enhancing children’s muscular fitness. The findings indicated that exercise interventions, predominantly combined interventions targeting selected domains of muscular fitness, were associated with significant moderate increases in local muscular endurance, muscular strength, and muscular power, with more significant effects observed with three sessions per week. Lastly, Zhao et al. [17] implemented a comprehensive strength training program for Chinese male adolescents to examine its effects on muscular fitness and perceived physical competence. The comprehensive strength training intervention improved their muscular fitness, notably lowered their extremity muscle power and abdominal core endurance, and increased their perceived physical competence, according to the study.

According to the studies mentioned above, strength training programs, especially those used in educational settings, can enhance children’s and teenagers’ perceived physical competence, as well as their muscle fitness. The studies also emphasize how crucial it is to take into account the type of resistance training intervention being utilized, as higher-repetition moderate-load regimens may be more successful in enhancing muscle endurance. We adhered to these recommendations while designing specific circuit training programs to improve local muscular endurance. The benefit of circuit training programs involving body weight, resistance bands, and medicine ball exercises in enhancing local muscular endurance in elementary school pupils, however, is not supported by sufficient research.

While previous studies incorporated school-based circuit training to improve local muscular endurance, there is insufficient evidence on how individuals’ baseline local muscular endurance fitness may affect the effectiveness of the applied treatment. Therefore, this study considered the individual baseline local muscular endurance capacity when exploring the effects of the applied treatment, which could provide valuable information for practitioners and educators designing school-based physical education programs. Additionally, this study may contribute to the literature by investigating the effectiveness of a circuit training program that uses body weight, resistance bands, and medicine ball exercises, specifically for school-based programs, in improving local muscular endurance in primary school boys, which does not interfere with the other contents of the PE curriculum. The main idea was to include local muscular endurance training in the first 15 min of a PE class to produce significant effects, which aligns with the previous study by Faigenbaum et al. [18]. Therefore, creating a strength training program that relied on limited resources and was space and time efficient, without interfering with the prescribed contents of the physical education curriculum, was desirable.

Based on the aforementioned, the purpose of this study was to determine whether a 12-week circuit training program that uses body weight, resistance bands, and medicine ball exercises, designed especially for school-based programs, can increase local muscular endurance in primary school boys who are of normal weight, while taking into account each participant’s individual baseline local muscular endurance capacity. Additionally, the study aims to provide valuable information for practitioners and educators designing school-based physical education programs and contribute to the literature on the effectiveness of school-based exercise interventions for enhancing primary school students’ muscular fitness.

## 2. Materials and Methods

### 2.1. Study Design

This parallel-group study compared the effects of circuit training with the conventional physical education curriculum (allocation ratio, 1:1). For our study in urban school settings, we selected four elementary schools from Belgrade and performed both the experimental and control treatments, as well as the outcome measurements, in school gymnasiums. Normal-weighted boys aged 11–14 followed one of the treatments, during the first school semester from October to December. After 12 weeks of treatment, the participants underwent a final measurement to assess whether any improvements had emerged to the response variables (sit-ups, DTE, and push-ups). Our study was approved by the Faculty of Sport and Physical Education’s ethical council and was carried out using experimental methods that adhered to the Helsinki Declaration’s ethical norms. Before conducting the baseline measurements, parental/legal guardian written consent forms were collected. No further modifications to the design emerged following the study’s commencement.

### 2.2. Participants

Six hundred and fifty primary school boys participated in this parallel-group randomized trial. The study inclusion criteria were: (1) elementary school boys aged 11–14, (2) BMI values within acceptable normal ranges for the age indicated in the classification proposed by Onis et al. [19], (3) they did not suffer from chronic diseases, and (4) they were not involved in any organized form of physical activity for at least three months prior to the start of the trial. Participants with BMI values outside of acceptable normal ranges, respiratory and cardiovascular diseases, developmental disabilities, students who were recovering from an injury or illness based on a recent medical diagnosis and receiving prescribed medical treatment, and participants who were engaged in organized physical activity at the time of the randomization were all excluded. We excluded participants who did not adhere to the experimental and control protocols ≥ 85%. A purpose-designed questionnaire evaluated the eligibility criteria for inclusion in the study, while Figure 1 shows the student selection process.

### 2.3. Physical Education Group Interventions

The total duration of the program was 12 weeks, with a frequency of 3 45 min physical education classes (total number of sessions = 36). The structure of the physical education classes consisted of the preparatory (15 min), main (25 min), and cooldown phase (5 min). Each physical education class began with a 10 min low-intensity warm-up period with brisk walking and light running alternations, followed by 5 min of whole-body mobility exercises. The main part of the classes contained elements of various sports (athletics, gymnastics, swimming, futsal, basketball, volleyball, and handball) and exercises designed to improve strength, muscular and cardiovascular endurance, speed, agility, stability, flexibility, and coordination. We should note that common strength-based strategies in regular PE curriculums in Serbia primarily consist of traditional exercises like push-ups, sit-ups, trunk extensions, and squats, which could ultimately contribute to the overall effectiveness of the PE curriculum on local muscular endurance. The only difference was that higher grades had more complex technical elements than lower grades. In general, we adopted the program of the Ministry of Education of the Republic of Serbia.

### 2.4. Circuit Training Group Interventions

We integrated the circuit training program into the main part of the physical education class and implemented three weekly resistance workouts according to the recommendation for novices, since the rest days between the sessions allowed for adequate recovery [10,11,12].

We adopted circuit training due to the limited time and the substantial number of participants in physical education classes, to keep the class active. Therefore, we implemented circuit training where the order of exercises topographically alternated between upper- and lower-body exercises, which allowed the individuals to recuperate more adequately between sets. This approach proved advantageous for untrained individuals who may have found it too taxing to perform multiple upper- or lower-body exercises consecutively. Additionally, this method of exercise arrangement decreased the duration of the rest periods between the exercises and maximized the rest between body areas [14]. This study’s local muscular endurance training program had brief rest periods, no more than 45 s, followed by an optimal number of repetitions performed with the participant’s own body weight or light loads. In addition, due to the lack of sophisticated equipment, we used the rate of perceived exertion for children scale to control the targeted session intensity [20].

We were fully aware of the advantages and disadvantages of the circuit training approach before determining the exercise plan. For instance, since bodyweight resistance training is limited to the individual’s body weight, the intensity might alter when increasing the number of repetitions or changing the movement pattern. Therefore, we implemented effort intervals of no less than 30 s. The advantage of such a specific approach is that the teacher might manipulate the load by adjusting the effort interval and exercise tempo. However, from a practical standpoint, we were aware that some individuals could not perform exercises in a given repetition range due to a lack of fitness or other common restrictions, like a range of motion issues, that could impair correct exercise posture. Therefore, if the individual could not perform at least 12 repetitions of a given exercise while maintaining the correct exercise posture, a more accessible alternative (modified exercise) or assistance was implemented. Each exercise contained three sets, and the adopted exercise tempo was 1-0-1-0 with some exceptions (weeks 9–11), where the exercise tempo was 2-0-2-0. The reason for such alteration was to maintain an optimal number of repetitions per set as the work interval rises and to employ a slightly different stimulus. Additionally, the participants were allowed to rest briefly before continuing the set to maintain the proper exercise posture.

To efficiently stress the skeletal muscle system, the participant was required to maintain a stable body position that permitted a safe and correct body alignment throughout the activity. During unsupported ground-based workouts, the feet were broader than shoulder-width apart. The experimental program primarily consisted of exercises to isolate specific abdominal muscles and large muscle groups. For example, students performed prone, supine, and side planks to isolate the abdominal musculature. Moreover, students used elastic bands and medicine balls as training aids (see Table 1 for more detail).

### 2.5. Baseline and Final Assessment

Before randomization, we performed the anthropometric measurements following the recommendations of Eston and Reilly [21]. They included measurements of body height, body mass, and lean body mass (LBM). Body height was measured with an anthropometer according to Martin (GPM, Switzerland) (measurement accuracy 0.1 cm), and body mass was measured to the nearest 0.5 kg using a portable electronic digital scale (Tanita UM-72, Made in Japan). We used the equation LBM (kg) = (body weight in kg) × (1 − (body fat %/100)), where the body fat percentage was calculated using the two-site skinfold measuring method (triceps and subscapular) and the formula provided by Slaughter et al. [22]. If participants met all the inclusion criteria, they underwent the baseline measurement procedures. For the baseline and final time points, we conducted test sessions simultaneously to avoid fluctuations due to diurnal changes in performance. All testing procedures were performed at school gyms to keep the testing surfaces and climate constant.

Previously physically taxing exertion by participants before testing led to exclusion. Before the baseline assessment, we advised participants to hydrate regularly and not be subject to diet restrictions. The standardized testing procedures also entailed refraining from taking supplements or medication before the test. The warm-up procedure was standardized and comprised of a general dynamic warm-up, such as jogging or light calisthenics (5 min), and a specific warm-up incorporating movements similar to those required by the test at moderate intensity (5 min). Familiarization with the tests began at the start of the school semester, lasting one month; however, they occurred without a structured program that could impact the results of this study. The tests on local muscle endurance were conducted continuously without the benefit of rest intervals and excessive body movements. The same three raters administrated each test, and the interrater agreement on the final score had to be absolute, or the participant had to repeat the test. The rest interval between the tests and each repetition was five minutes, according to the recommendations by McGuigan [23]. During the rest period, we advised the participants to walk around to accelerate recovery. As for the local muscular endurance tests, the testing procedure included accomplishing the maximum number of sit-ups, push-ups, and dynamic trunk extension (DTE) repetitions.

### 2.6. Sit-Up Test (30 s)

The participants lay on a flat, soft surface (judo mat) with their legs bent at the knees at 90 degrees, palms crossed at the back of their head, and elbows pointing aside, while touching the floor with their elbows during the descending part of the movement and touching their knees during the ascending part. The tester assisted the participants by fixing their feet to the ground and controlling the position and angle of the feet and knees. The testing procedure lasted 30 s, while two testers controlled the time with a stopwatch for accuracy. If the participant did not touch the mat or knees with their elbows, the repetition did not count. The intraclass correlation coefficients for the test–retest reliability and typical error for the sit-up test in this study were 0.97 and 1.44%, respectively.

### 2.7. Parallel Roman Bench Dynamic Trunk Extension (DTE) Test

This test required the subject to perform the maximum number of repetitions of the trunk muscle extension on a Roman bench, with an accompanying sound repeated every 3 s [24]. The test ended when the subject could not perform more than 20 repetitions per minute or voluntarily abandoned testing. We used an adjustable parallel Roman bench (height: 83.82 cm; length: 114.3 cm; width: 60.96 cm) to account for the variety of body sizes. The participants took the neutral starting position on the Roman bench with arms crossed on their chest, while their legs and trunk were in the same plane. From this position, the participant bent forward until he achieved a 90-degree angle between the torso and legs. Each repetition matched the audio signal repeated every 3 s, and after every repetition, the subject returned to the starting position. Three testers administrated this test, to guarantee the subject’s safety and the measurement’s accuracy. Only correctly performed attempts, in each minute, counted. The intraclass correlation coefficients for the test–retest reliability and typical error for the DTE in this study were 0.97 and 2.52%, respectively.

### 2.8. Push-Up Test (30 s)

The 30 s push-up test aimed to assess the local muscular endurance of the upper body. The participants adopted a prone position with their hands placed shoulder-width apart, feet together, and fingers pointing forward. The participant pushed upwards to full arm extension, roughly forming a straight line from the shoulders to the ankle. By flexing their arms, the participant descended the body to a posture where the chest-to-thigh body area touched the ground. The participants maintained a straight body position without swaying for the test duration. The failure to maintain a straight body position or attain the upward or downward position resulted in no count. During a 30 s interval, we recorded the correct repetitions while two testers controlled the time with a stopwatch for accuracy. The intraclass correlation coefficients for the test–retest reliability and typical error for the push-ups in this study were 0.97 and 1.30%, respectively.

### 2.9. Randomization and Blinding

We created randomization schemes for each stratum (fifth, sixth, seventh, and eighth grade), printed each on a card, sealed it in a secure envelope with the participant ID, and provided equal numbers to four schools. Moreover, we expanded the randomization list of assignments due to potential dropouts and uneven enrolment. We used a permuted block design with block sizes of 2, 4, and 6 to prevent a predictable pattern in the randomization. The number of potential participants for each stratum was proportional to the total number of students in each grade, namely 20%, 30.8%, 30.8%, and 18.4% for the fifth, sixth, seventh, and eighth grades, respectively. After checking all the inclusion criteria during the anthropometric measurements and the baseline assessment, trained personnel assigned the participants to the experimental or control group. Given the nature of the trial, we did not blind the personnel, physical education teachers, and participants to the exercise allocation given the nature of the trial. However, the administrators of the baseline and final measurement tests were blinded.

### 2.10. Sample Size Calculation

We conducted an a priori power analysis using G*Power [25] for the three-way mixed model ANOVA with eight groups (treatment (two levels: experimental and control), and grade (four levels: fifth, sixth, seventh, and eighth)) and two measurements (baseline and final) as the input parameters. We also performed a priori analysis for the two-way between-subject ANCOVA (same number of groups), with the numerator degrees of freedom value of three for the interaction effect. Given values of alpha (0.05), power (0.80), and expected small effect size (f = 0.14) were the parameters of choice for the sample size calculations. Based on these assumptions, the desired sample size for this study was 256 and 539 participants for ANOVA and ANCOVA, respectively.

### 2.11. Statistical Analyses

Descriptive statistics were calculated to summarize the data collected in this study. Continuous variables’ measures of mean and standard deviation were calculated. A three-way ANOVA (treatment (experimental, control), grade (four, levels: fifth, sixth, seventh, eighth), and timepoint (baseline, final)) was performed to determine the interaction effect between the independent variables. We converted a vector of *p*-values into a letter summary using a character-based display that utilized familiar alphabetic characters. The letter summary was used to indicate the similarities and differences among the groups or levels that were not statistically significant [26]. A two-way analysis of covariance (ANCOVA) determined the effects of the experimental treatment. The baseline measurement values were used as the covariate to adjust the mean scores at the final measurement. Post-hoc pairwise comparisons were conducted using the Bonferroni correction to adjust for multiple comparisons. The effect size criteria were: <0.2 trivial effects, 0.2–0.6 small effects, 0.6–1.2 moderate effects, 1.2–2.0 large effects, and >2.0 very large effects [27]. The data were processed using RStudio (version 2022.07.0.548, Spotted Wakerobin, Boston, MA, USA). We set the statistical significance level at *p* < 0.05.

## 3. Results

We examined the study outcomes for 606 participants. In more detail, the exclusion criteria during the follow-up prior to data analyses are presented in Figure 1. Additionally, data screening was essential to detect outliers and check for assumptions. Twenty-eight multivariate outliers were found using the Mahalanobis distance as a criterion and excluded from the analysis (*n* = 578 included in the final analysis). The data met the multivariate assumptions of normality, linearity, homogeneity, and homoscedasticity. After conducting anthropometric measurements, including body height (BH), body mass (BM), body mass index (BMI), and lean body mass (LBM), we observed no significant differences between the baseline and final time points for students in the fifth, sixth, seventh, and eighth grades (*p* > 0.05) (see Table 2.).

The mixed model ANOVA indicated significant improvements in the subjects for sit-ups (F = 9.89, *p* = 0.002, *η_p_^2^* = 0.01), DTE (F = 5.71, *p* = 0.017, *η_p_^2^* = 0.003), and push-ups (F = 4.63, *p* = 0.032, *η_p_^2^* = 0.004). Both the experimental and control groups exhibited significant improvements. The effect sizes for sit-ups, DTE, and push-ups were 0.42, 0.33, and 0.40 for the experimental group and 0.49, 0.18, and 0.33 for the control group, respectively.

Additionally, there was a significant interaction effect between the grade and type of treatment (experimental and control) for sit-ups (*F* = 6.56, *p* < 0.001, *η_p_*^2^ = 0.02) and DTE (*F* = 11.36, *p* < 0.001, *η_p_*^2^ = 0.03).

Conversely, when we examined the mean individual sit-up outcome, we saw a completely different pattern. The responses in fifth graders were noticeably higher in the control group (see Figure 2). However, the seventh and eighth grade responses were significantly higher for the experimental group, while there were no significant alterations between treatments for the seventh grade.

Two example models were evaluated to assess more appropriate methods for predicting sit-up repetitions: ANOVA and ANCOVA (see Figure 3). The results showed that the R^2^ for the ANOVA model was 0.11, indicating that the model explained only 11% of the variance in the sit-up repetitions, while the ANCOVA model explained 95% of the variance in the sit-up repetitions (R^2^ = 0.95). The ANCOVA model’s high R^2^ and slope values suggest that it was appropriate for predicting the actual sit-up repetitions. A similar trend was observed for DTE and push-ups.

The baseline measurement scores were a significant adjustor for all the response variables (sit-ups, DTE, push-ups, and body fat), indicating that the baseline physical fitness contributes to the overall treatment outcome.

After adjusting for the baseline measurement, the interaction effect between the treatment and grade when accounting for the covariate was significant for sit-ups (*F* = 7.74, *p* < 0.001, *η_p_*^2^ = 0.04), DTE (*F* = 6.49, *p* < 0.001, *η_p_*^2^ = 0.03), and push-ups (*F* = 9.22, *p* < 0.001, *η_p_*^2^ = 0.05) (see Figure 4).

## 4. Discussion

This study aimed to explore the effects of moderate to intense school-based circuit training on local muscle endurance. The study’s findings verified that a 12-week circuit improved the local muscular endurance in normal-weighted primary school boys. All the response variables (sit-ups, DTE, and push-ups) demonstrated significant within-subject improvements for the experimental and control groups. For sit-ups, DTE, and push-ups, the experimental group improved by 7.19, 8.71, and 13.27% (Es = 0.42, 0.33, and 0.40), respectively, while the control group improved by 3.45, 6.41, and 7.83% (Es = 0.49, 0.18, and 0.33), respectively. These improvements are slightly less beneficial than those of one previous study regarding local muscular endurance, where the effect sizes were 0.61 and 0.67 for DTE and dynamic curl-ups, respectively [28]. The systematic review and meta-analysis by Villa-González, Barranco-Ruiz, García-Hermoso, and Faigenbaum [16] found that school-based exercise interventions were associated with significant moderate increases in local muscular endurance (g  =  0.65 95% CI, 0.13 to 1.17, *p*  =  0.020; I^2^  =  85.0%), with higher effects when using interventions with ≥3 sessions per week. Zhao, Liu, Han, Li, Liu, Chen, and Li [17] proved that school-based interventions improved local muscular endurance in sit-ups (14.96%) and push-ups (15.73%), the finds of which are somewhat similar to our study.

Additionally, we should note that one of the crucial goals of the physical education curriculum in Serbia is to improve local muscular endurance, and the most applied exercises to improve that trait are sit-ups and push-ups. It is uncertain whether a substantial training effect in the control group was related to gains in local muscular endurance, increased familiarity with the assessment items, or higher motor skill performance during movement execution. Therefore, considering that our experimental group did not employ test-specific exercises, the findings of this study provide evidence that the increases in local muscular endurance were due to the experimental treatment itself. The overall improvements for both the experimental and control groups were small in magnitude, but not negligible. Additionally, the experimental group benefited more from grade-based changes. For example, the gains in DTE for the fifth, sixth, seventh, and eighth grades were 9.74, 7.18, 7.38, and 5.18% (experimental group) vs. 2.52, 4.69, 2.13, and 4.02% (control group), respectively. The interaction impact of specific outcome variables significantly alternated between the experimental and control groups, according to pairwise analyses of the simple main effects by grade (see Figure 2). For instance, the fifth grade showed the most significant mean difference between the control and experimental groups, with mean individual responses for DTE being more favorable in the control group (fifth, sixth, and seventh grade).

In contrast, the eighth grade presented no alterations across treatments, meaning the grade-based responses did not have a similar pattern. Interestingly, the eighth grade responses were significantly lower than sixth grade in the control group. Generally, the effects of the applied treatments were too mixed and complex to explain clearly. Therefore, we used the significant main effects of the treatment and grade and treatment–grade interaction to build the multiple regression model for each response variable to explore the individual contribution of the predictors and their interaction. The multiple R-squared coefficients were low, and the grade had the most significant contribution to the model, while the contribution of the treatment and grade–treatment interaction was significant, however small (results not presented).

However, when we introduced the baseline scores for the response variables to the model (ANCOVA model), we obtained a more suitable model to predict the effects of the applied treatments (see the example in Figure 3). Consequently, conducting a two-way between-subject ANCOVA was necessary to explore the meaningful effect of the applied treatments when accounting for grade and their mutual interaction. The ANCOVA was used to analyze the effect of the applied treatments and the grade–treatment interaction on the response variables (sit-ups, DTE, and push-ups), after adjusting for the baseline measurement scores.

When we observe the pairwise comparisons of the simple main effects adjusted for the baseline measurement to improve the inference in the experimental treatment factor, it is apparent that the interaction effect shows a significant difference between the treatments in favor of the experimental group (see Figure 4). The slopes in the regression lines for the control and experimental groups are different for the conditional expectations at a specific covariate value (the baseline measurement score). One could observe that the effect of the experimental treatment is not constant for all the baseline values compared to the control. As the baseline values increase, the effects of the implemented experimental treatment become smaller, and a similar trend is present for all the response variables. Likewise, the baseline to the final score’s effect depends on the treatment type, where the effect of the control group is significantly smaller than the experimental group, with some respected exceptions in the simple main effects (see Figure 4 for pairwise comparisons).

Additionally, by inspecting Figure 4, we can observe that the grade responses replicate the pattern of all the response variables. For example, the fifth and sixth grade responses are approximately similar to or higher than others at smaller values in the baseline measurement. However, as the baseline values increase, the grade’s effects become smaller. Likewise, the baseline to final score effect depends on the grade, where the treatment effect is more beneficial for the lower grades in the experimental group, at least at the numerical level. Conversely, the effects of the control treatment do not follow a similar pattern (see Figure 4 for pairwise comparisons).

Moreover, previous studies proved that relative strength improvements in prepubescents are equivalent to or larger than in adolescents, consistent with our results [9,13]. For younger children, the experimental treatment was more effective. For instance, push-up treatment effects were more significant in favor of the experimental group among fifth graders than older students. Similarly, grade-based comparisons within the experimental group demonstrated that fifth graders gain more than older students from the applied treatment (see Figure 4). Another possible explanation for such outcomes is that PE teachers do not conduct the PE curriculum in Serbia before the fifth grade, which could impact the children’s overall attenuated physical fitness. Therefore, there is a possibility that the experimental treatment proved more effective for fifth graders based on this assumption.

In contrast, the outcomes of the control group did not exhibit the same pattern, indicating that the effects of physical education were not significantly different between grades for push-ups. However, the experimental treatment effects differ based on an individual’s baseline local muscular endurance capacity. As baseline levels grow, the treatment and grade impacts become less favorable (see Figure 4). The variability in strength increase may be attributable to several variables, including the individual’s biological age, the program design, the instructional quality, and past training experience. Therefore, we cannot fully explain this phenomenon, but we assume that stronger individuals benefit less from the treatment due to insufficient stimulus, which may even induce detraining effects [29]. Nevertheless, scientific research indicates that children may significantly enhance their strength, regardless of their development and maturity, when provided with an appropriate resistance training program [10]. Therefore, a well-designed circuit training plan for each individual may induce more significant improvements.

Furthermore, neural changes are essential mechanisms for improvements in strength. They may be a viable explanation for local muscle endurance, mainly when, hypothetically, we could not observe substantial alterations in lean body mass, which might indicate that 12-week school-based local muscular endurance did not affect muscular hypertrophy. The possibility of prepubescent individuals experiencing alterations in muscular hypertrophy through resistance training has been debated. Some authors argue that inadequate circulating hormone levels may prevent significant changes in muscle size without at least 20 weeks of training [13]. However, the efficacy of resistance training in promoting muscular hypertrophy in this population cannot be confidently determined without advanced analysis.

Additionally, Lloyd and Faigenbaum [13] argue that due to neurological characteristics, such as motor unit activation and synchronization improvements, increased motor unit recruitment, and firing frequency, pre-adolescents have more potential for strength gain. The same authors emphasize that changes in recruitment patterns, muscle protein, and connective tissue may occur in children’s muscles and nervous systems. Additionally, they conclude that intrinsic muscle adaptations, increases in motor skill performance, and the synchronization of the implicated muscle groups may contribute to training-induced strength gains in prepubescents. The statements above could further explain the positive effect of the control group by improving movement patterns similar to the testing procedures.

When we considered implementing circuit training, we relied on the suggestions of Fleck and Kraemer [10] that lighter loads (12RM and lighter) could improve local muscular endurance. Additionally, students without experience of strength training, performing 10 to 15 light resistance repetitions in each set, may experience local muscular endurance similar to higher training intensity [11,30,31]. For example, using larger loads (60 to 80% of 1RM) and including short rest intervals and several sets aids in increasing muscle endurance [12]. However, this strategy may be time-consuming and be difficult to adopt in a typical school setting, especially for students that lack proper lifting techniques and in schools with limited resources. In contrast, low-intensity local muscular endurance may improve by performing more repetitions and utilizing sets with significantly reduced weights [12]. Arguably, resistance training to improve local muscular endurance may alter an individual’s capacity to sustain a particular number of repetitions per set with no change in resistance or lead to smaller reductions in the resistance required to maintain the same number of repetitions. This approach allows for the manipulation of the work-to-rest ratio and exercise tempo to introduce a different stimulus for the student to adapt and maintain the prescribed repetition range (see Table 1).

Moreover, beginners should follow a 2 to 3 weekly whole-body program to enhance local muscular endurance, utilizing gradual weekly increments of 5 to 10 percent [9,11]. Consequently, a program for children and adolescents may consist of 10 to 15 repetitions in each set, including at least one exercise for each of the body’s major muscle groups [9,10,11,12]. Zatsiorsky, Kraemer, and Fry [12] suggest that the training intensity could be altered by adjusting the number of repetitions, sets, or exercises in each training session and the duration of the rest intervals between exercises, which seems applicable in physical education classes. We ensured that the number of repetitions in a single set did not surpass the point of failure. According to Fleck and Kraemer [10], training to failure has not been demonstrated to increase local muscle endurance. Multi-joint, whole-body workouts may implement this type of training as well.

Our premise was to implement a whole-body workout regimen to target large muscle groups with available school equipment like resistance bands and medicine balls. In addition, a teacher must consider the developmental variations among children of the same age when designing a training program [32]. Physically and mentally, pre-adolescents and adolescents of the same age vary. Genetic and development rate disparities result in physical and psychological differences. Children should begin resistance training at a level suitable to their development, physical ability, and personal objectives [9]. When introducing children to resistance training, it is always preferable to underestimate their physical ability and gradually raise the volume and intensity of the training rather than surpassing their limits and risking injury or long-term adverse health effects [13].

Understanding fundamental growth and development concepts can aid in creating applicable resistance training programs for children and adolescents in the school setting. This comprehension will also aid in developing resistance training program objectives and exercise progression [10]. Safety and injury prevention is highly significant when contemplating resistance training for children and adolescents, and most school-reported injuries may result from inadequate supervision, poor technique, or competitive factors [8]. Resistance training programs are comparatively risk-free compared to other sports and activities in which children and adolescents routinely engage, which proved to be the case in our study because the adherence rate was reasonably high (see Figure 1). Contrariwise, it seems that the stresses exerted on the joints of young athletes during sports participation may be much larger and more challenging to predict than those produced by resistance training regimens [30].

Therefore, circuit training seems an efficient and practical method for enhancing local muscular endurance. We knew that circuit training is often not suggested because of the modest strength improvements. However, we can implement it when strength is not the most crucial motor skill, such as in the physical education curriculum. Circuit training seems feasible to achieve optimal results when improving local muscular endurance, where body weight, resistance band, and medicine ball exercises may be performed [12]. This approach is rather practical and meant to be time efficient, considering that the short duration of the physical education class does not leave much room to implement other vital contents, such as skills acquisition in various sports.

Comprehensive school-based programs should improve health-related fitness components, such as muscular strength and endurance, cardiorespiratory endurance, flexibility, and body composition. Smith, Eather, Morgan, Plotnikoff, Faigenbaum, and Lubans [5] offer evidence for the existing physical activity guidelines, which encourage children and adolescents to exercise regularly using muscle-strengthening activities. The authors argue that in addition to other health- and skills-related components of physical fitness, school-based youth programs should incorporate exercises that improve muscle strength, local muscular endurance, and muscular power. Physical education may be the only time and location throughout the school day when every child and adolescent may acquire information and skills connected with physical exercise and engage in physical activity. The regulated and supervised educational setting is the only time and location where all children can safely participate in moderate or vigorous physical exercise [33].

Additionally, based on their behavior, most children and adolescents fail to participate in moderate- to vigorous-intensity physical exercise for the recommended 60 min or more per day, with up to one-third reporting that they do not partake in any physical activity [1]. This lack of physical exercise has increased childhood obesity and decreased fitness. Therefore, the physical education curriculum should be founded on an awareness of the growth patterns and developmental phases and provide sound strategies to increase physical activity to develop motor skills via an appropriate movement strategies approach [1,33].

Finally, we should point out that the current study had several limitations, including: (1) the lack of non-exercise activity monitoring systems, (2) there was no assessment of the treatment transfer to other motor abilities, (3) there was no implementation of intensity tracking devices, and (4) the generalizability of the study findings was limited due to the exclusion of specific populations, such as female participants and overweight/obese individuals. However, apart from a few limitations to the present study, the experimental treatment’s strength is that it is efficient, time saving, reproducible, and easy to manage, even in environments with limited resources, which may be valuable when taking into account the practical implications.

## 5. Conclusions

The study’s results confirmed that a 12-week circuit training program increased the local muscular endurance in normal-weighted primary school boys. All response variables (sit-ups, DTE, and push-ups) showed substantial within-subject improvements in the experimental and control groups. However, the experimental treatment was more beneficial than the control and varied depending on an individual’s baseline local muscular endurance capacity. As baseline levels increase, the treatment and grade effects become less beneficial. We suggest that each individual’s circuit training program can be tailored to their personal needs to maximize the gains. This training method may be applied as multi-joint, total-body workouts and possibly produce optimal results using body weight, resistance bands, and medicine ball exercises. The use of body weight, resistance bands, and medicine ball exercises with this specific training strategy may result in multi-joint, total-body workouts that are effective and easily implemented in school settings. Therefore, the practical implications of this study may be valuable. Nevertheless, as we did not assess how the treatment impacts girls or overweight and obese students, the findings of this trial cannot be generalized to the overall primary school population.

In addition, comprehensive school-based programs should increase local muscular endurance, as a part of health-related fitness. Nowadays, most children and adolescents do not engage in moderate- to vigorous-intensity physical activity for at least 60 min per day, as is advised. Therefore, the physical education curriculum should rely on understanding the growth patterns and developmental stages during which local muscular endurance circuit training may be advantageous. Future studies should explore whether long-term local muscular endurance circuit training could improve health-related fitness using more sophisticated monitoring devices.

## Figures and Tables

**Figure 1 children-10-00726-f001:**
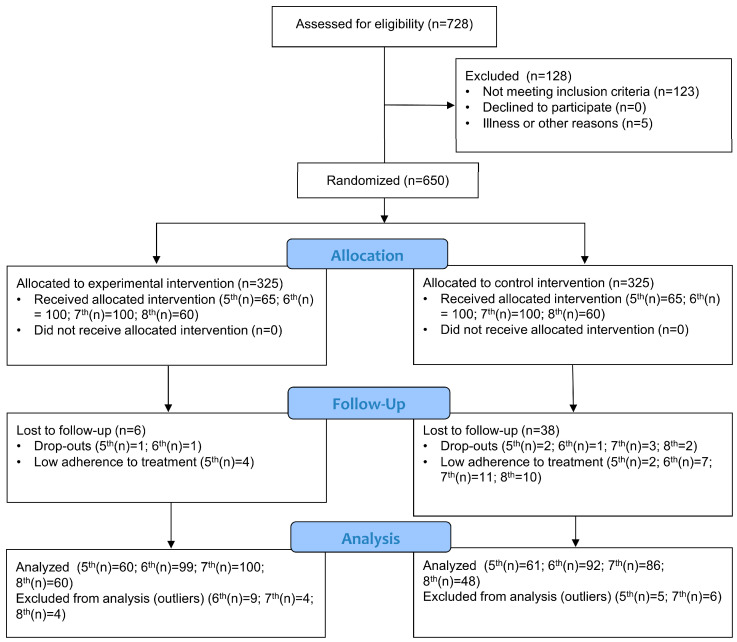
Flow chart diagram of subjects included in the study, randomization, and analysis.

**Figure 2 children-10-00726-f002:**
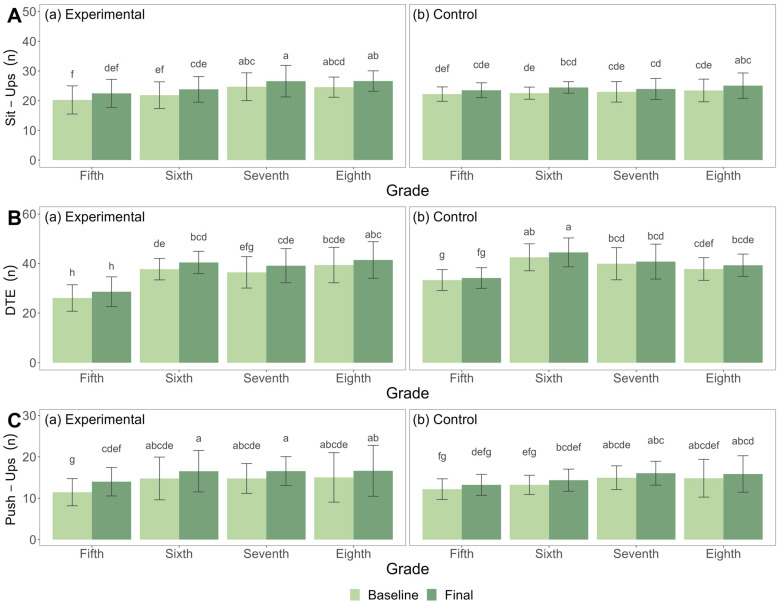
Mixed model (2 × 2 × 4) three-way ANOVA showing the effect of the group (experimental and control; panels (a) and (b)), time point (baseline and final), and grade (fifth, sixth, seventh, eighth) on sit-up, DTE, and push-up performance. (**A**–**C**) represent the main and interaction effects for sit-ups, DTE, and push-ups, respectively. We used alphabetic characters to convert a vector of *p*-values into a letter summary to indicate significantly different means. Based on a post-hoc Tukey’s HSD test, the letters above the bars indicate significant the differences between the groups, time points, or grades (*p* < 0.05). The means with at least one common letter are not significantly different. For example, the means for the fifth grade in the experimental group (baseline vs. final) may seem significantly different based on the contrasting letter labels (f vs. def). The means are different, however not significantly, because they share a common letter f. The bar plots display the group mean and error bars as standard deviation.

**Figure 3 children-10-00726-f003:**
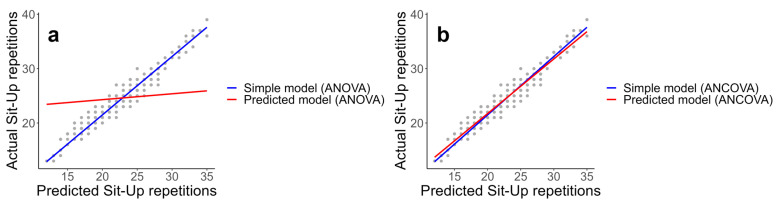
Example linear regression fitted to actual and predicted sit-up repetitions. The R^2^ value was 0.11 and 0.95 for ANOVA (**a**), and ANCOVA (**b**) models, respectively. In this example, the high R^2^ and slope values suggest that the ANCOVA model was appropriate for predicting the actual sit-up repetitions.

**Figure 4 children-10-00726-f004:**
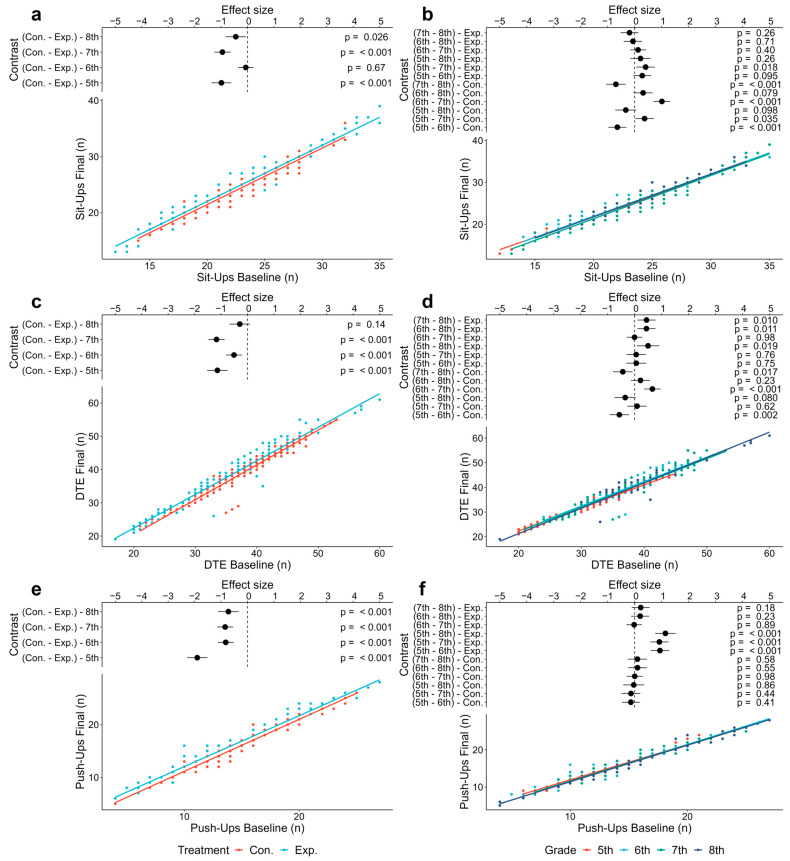
The two-way ANCOVA interaction plot shows the treatment effect (experimental vs. control) on the outcome of the response variables, adjusted for the baseline measurement. (**a**,**c**,**e**) represent the interaction effects (grade presented) for sit-ups, DTE, and push-ups, respectively. (**b**,**d**,**f**) represent the interaction effects (treatment presented) for sit-ups, DTE, and push-ups, respectively. The regression lines with data points display the treatment–grade interaction, while the effect size and *p*-value represent the simple main effects.

**Table 1 children-10-00726-t001:** Circuit training program.

**Session 1**				
**Exercise**	**Slide Board Split Squat**	**Single-Leg Hip Thrust**	**Prone Plank**	**Gymnastics Arabesque Hold (Left and Right)**
Week	Work: Rest/RPE	Work: Rest/RPE	Work: Rest/RPE	Work: Rest/RPE
Week 1–4	30″ by 45″/6	30″ by 45″/6	30″ by 45″/6	30″ by 45″/6
Week 5	35″ by 40″/6	35″ by 40″/7	35″ by 40″/7	35″ by 40″/7
Week 6	40″ by 35″/7	40″ by 35″/7	40″ by 35″/7	40″ by 35″/7
Week 7	45″ by 35″/7	45″ by 35″/7	45″ by 35″/7	45″ by 35″/7
Week 8 (deload)	35″ by 40″/6	35″ by 40″/6	35″ by 40″/6	35″ by 40″/6
Week 9	45″ by 35″/7	45″ by 35″/7	45″ by 35″/7	45″ by 35″/7
Week 10	50″ by 30″/8	50″ by 30″/8	50″ by 30″/8	50″ by 30″/8
Week 11	55″ by 30″/8	55″ by 30″/8	55″ by 30″/8	55″ by 30″/8
Week 12 (deload)	45″ by 30″/7	45″ by 30″/7	45″ by 30″/7	45″ by 30″/7
**Session 2**				
**Exercise**	**Alternating single-leg Romanian deadlift (medicine ball)**	**Knee push-ups**	**Prone plank (alternating arm and leg raise)**	**Supine plank (hands on a medicine ball)**
Week	Work: Rest/RPE	Work: Rest/RPE	Work: Rest/RPE	Work: Rest/RPE
Week 1–4	30″ by 45″/6	30″ by 45″/6	30″ by 45″/6	30″ by 45″/6
Week 5	35″ by 40″/7	35″ by 40″/7	35″ by 40″/7	35″ by 40″/7
Week 6	40″ by 35″/7	40″ by 35″/7	40″ by 35″/7	40″ by 35″/7
Week 7	45″ by 35″/7	45″ by 35″/7	45″ by 35″/7	45″ by 35″/7
Week 8 (deload)	35″ by 40″/6	35″ by 40″/6	35″ by 40″/6	35″ by 40″/6
Week 9	45″ by 35″/7	45″ by 35″/7	45″ by 35″/7	45″ by 35″/7
Week 10	50″ by 30″/8	50″ by 30″/8	50″ by 30″/8	50″ by 30″/8
Week 11	55″ by 30″/8	55″ by 30″/8	55″ by 30″/8	55″ by 30″/8
Week 12 (deload)	45″ by 30″/7	45″ by 30″/7	45″ by 30″/7	45″ by 30″/7
**Session 3**				
**Exercise**	**Bulgarian split squat (medicine ball on chest)**	**Deadlift with resistance bands**	**Side plank (both sides)**	**Split-stance shoulder press with resistance band**
Week	Work: Rest/RPE	Work: Rest/RPE	Work: Rest/RPE	Work: Rest/RPE
Week 1–4	30″ by 45″/6	30″ by 45″/6	30″ by 45″/6	30″ by 45″/6
Week 5	35″ by 40″/7	35″ by 40″/7	35″ by 40″/7	35″ by 40″/7
Week 6	40″ by 35″/7	40″ by 35″/7	40″ by 35″/7	40″ by 35″/7
Week 7	45″ by 35″/7	45″ by 35″/7	45″ by 35″/7	45″ by 35″/7
Week 8 (deload)	35″ by 40″/6	35″ by 40″/6	35″ by 40″/6	35″ by 40″/6
Week 9	45″ by 35″/7	45″ by 35″/7	45″ by 35″/7	45″ by 35″/7
Week 10	50″ by 30″/8	50″ by 30″/8	50″ by 30″/8	50″ by 30″/8
Week 11	55″ by 30″/8	55″ by 30″/8	55″ by 30″/8	55″ by 30″/8
Week 12 (deload)	45″ by 30″/7	45″ by 30″/7	45″ by 30″/7	45″ by 30″/7

Note. Work: Rest—work-to-rest ratio expressed in seconds. Prescribed breaks express rest intervals between two consecutive exercises, not between sets. RPE—desired session rate of perceived exertion.

**Table 2 children-10-00726-t002:** Descriptive statistics.

**Experimental**								
		**Baseline**				**Final**		
**Grade**	**5th**	**6th**	**7th**	**8th**	**5th**	**6th**	**7th**	**8th**
BH [cm]	145.17 ± 5.56	159.14 ± 7.74	165.51 ± 7.85	176.70 ± 7.07	145.85 ± 5.44	159.73 ± 7.68	166.13 ± 7.81	177.12 ± 6.92
BM [kg]	35.96 ± 3.89	45.57 ± 6.74	52.42 ± 8.41	61.79 ± 8.92	36.35 ± 3.80	46.19 ± 6.79	53.02 ± 8.50	62.38 ± 9.00
BMI [kg/m^2^]	16.99 ± 0.90	17.92 ± 1.66	19.00 ± 1.62	19.71 ± 1.87	17.08 ± 0.89	18.03 ± 1.67	19.08 ± 1.62	19.80 ± 1.88
LBM [kg]	32.40 ± 3.44	39.61 ± 6.00	45.19 ± 7.76	53.91 ± 7.95	32.81 ± 3.34	40.24 ± 6.05	45.94 ± 7.91	54.66 ± 7.98
Sit-ups [*n*]	20.25 ± 4.74	21.75 ± 4.44	24.70 ± 4.68	24.55 ± 3.39	22.47 ± 4.72	23.82 ± 4.31	26.57 ± 5.32	26.61 ± 3.44
DTE [*n*]	26.08 ± 5.35	37.73 ± 4.33	36.45 ± 6.35	39.41 ± 7.15	28.62 ± 6.00	40.44 ± 4.51	39.14 ± 6.91	41.45 ± 7.38
Push-ups	11.45 ± 3.29	14.78 ± 5.17	14.78 ± 3.60	15.04 ± 5.98	14.00 ± 3.45	16.53 ± 5.01	16.55 ± 3.49	16.62 ± 6.15
**Control**								
		**Baseline**				**Final**		
**Grade**	**5th**	**6th**	**7th**	**8th**	**5th**	**6th**	**7th**	**8th**
BH [cm]	149.19 ± 8.00	157.65 ± 6.46	167.87 ± 6.92	173.95 ± 7.47	149.75 ± 7.85	158.29 ± 6.37	168.44 ± 6.72	174.29 ± 7.43
BM [kg]	37.98 ± 5.01	44.58 ± 6.50	54.81 ± 6.61	60.11 ± 7.50	38.45 ± 4.96	45.02 ± 6.45	55.20 ± 6.53	60.58 ± 7.58
BMI [kg/m^2^]	16.95 ± 0.98	17.90 ± 2.06	19.39 ± 1.31	19.81 ± 1.55	17.03 ± 0.99	17.93 ± 2.04	19.40 ± 1.29	19.89 ± 1.52
LBM [kg]	33.47 ± 4.44	38.82 ± 5.56	47.67 ± 6.09	52.45 ± 6.60	33.98 ± 4.40	39.31 ± 5.54	48.13 ± 6.03	53.17 ± 6.40
Sit-ups [*n*]	22.21 ± 2.42	22.57 ± 2.03	23.00 ± 3.43	23.44 ± 3.83	23.52 ± 2.50	24.48 ± 1.94	23.96 ± 3.51	25.06 ± 4.26
DTE [*n*]	33.32 ± 4.25	42.43 ± 5.41	39.96 ± 6.46	37.79 ± 4.58	34.16 ± 4.17	44.42 ± 5.80	40.81 ± 7.02	39.31 ± 4.52
Push-ups [*n*]	12.20 ± 2.49	13.27 ± 2.30	15.05 ± 2.99	14.83 ± 4.56	13.23 ± 2.54	14.41 ± 2.65	16.02 ± 2.87	15.85 ± 4.42

Note. BH—body height; BM—body mass; BMI—body mass index; LBM—lean body mass; DTE—Roman bench dynamic trunk extension.

## Data Availability

The authors will make accessible the raw data supporting the results of this research without excessive restraint.

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
