# Peer review of "School-Based Circuit Training Intervention Improves Local Muscular Endurance in Primary School Students: A Randomized Controlled Trial"

_children, 2023, doi:10.3390/children10040726_

Round 1

Reviewer 1 Report

Minor issue

- Generally, please summarize and shorten the introduction. It should be more concentrated on the intervention used in the study.

- Please replace “Sample Size” into the Participants.

- Please add descriptive statistics into the “statistical analyses”, too.

- Please mention clearly the practical implications of this study.

- References should be in accordance with the format of journal. Please correct all of them.

Reviewer 2 Report

Dear Authors

As one of the reviewers, I express my personal scientific opinion on your work. I would like to reassure you that I was trying to be positive and constructive but particularly as fair and honest as possible to your work. The high number of the sample size and the clear explanation provided in Method’s section is appreciated. I should also note that the originality of the study, the statistical approach used, the calculation of the Effect Size, the work done on figures and tables and the reporting of the limitations of the study are all positive points.

Please accept my judgment with a positive and constructive way.

General comments:

1.      I found the Introduction a bit exhaustive; 12 paragraphs introduction for a research article are too much. Please eliminate the generalizations within the Intro and keep concentrated in literature information which are highly relevant to your study subject for developing your research question.

2.      If it is possible, please restructure the whole intro in order to make it a hit to the point one.

3.      You are using the term “health” throughout the text but particularly in your Conclusion section. Which parameter, that is related to health, has been evaluated in your study?

4.      Although the article is in general understandable and well-presented, it is written in slightly an informal and not such an academic way. I would like to suggest you to revise the whole paper accordingly in order to meet the scientific criteria of written academic language.

Abstract:

5.      I would expect to see some numbers and/or percentages, and P values within the abstract.

Introduction:

6.      Lines 40-41: “However, whether … arises”. Arises what? Please check grammar for clarity.

7.      Lines 42-43: You wrote: “Physical activity recommendations… but have grown more specifically for children and adolescents”. Please check grammar for clarity. I would like to suggest you to re-write the whole sentence for making it clearer and understandable.

8.      I found the whole 2nd and 3rd paragraphs of the Introduction very basic and almost irrelevant with the background of the study. In my point of view, at the end of the 1st paragraph, you could write just a single sentence including the time usually spent as well as the traditional content included in the PE lessons/classes and then move straight forward to the 4th paragraph.

9.      Please check the whole Intro for grammar and syntax errors for improving the clarity of its content.

Methods:

10.  Why your study was focused only in boys; why you did not include girls in your experimental sample? What the girls were doing during the intervention application in boys?

11.  Lines 178-180: If a boy was not within the acceptable normal range for his age, as proposed by Onis et al., what was happening?

-          Did you exclude this particular boy from your study?

-          And if YES, why?

-          And if yes, what this boy was doing when the rest of his classmates were performing the circuit training intervention program?

12.  Line 182: You reported: “Exclusion criteria were BMI values above and below acceptable normal ranges”. Why did you set a such exclusion criterion; what is the physiological/biological explanation behind this exclusion criterion?

13.  Please report the post-hoc test you used into your Statistical analyses section.

14.  Please avoid reviewing the literature in your method section (i.e. lines 231-232; 305-306).

15.  Line 294: Please consider make it italics.

16.  Lines 351-353: The meaning of the sentence is not clear. To be honest I did not actually understand what you have done. Could you please re-write the whole sentence for giving a clearer meaning?

Results:

17.  You evaluated BMI and found no differences between groups. However, we are all aware that with 12 weeks of resistance training there is an increased also in muscle mass, and therefore an elevation of the weight of the muscles which is taken into consideration during the BMI calculation. Don’t you think that this may increase the Type II error? Why you did not evaluate body fat percentage just with skinfold caliper in order to potentially collect better results concerning lean muscle mass.

18.  Line 365: Did you want to report (n=578) and NOT n=579. In addition, please explain what is the n=578 since as it stands at the end of the sentence is confusing.

19.  Lines 373-375: Please consider to re-write the whole sentence.

20.  Lines 380-388: Please avoid discussing your results within the Results section. You could transfer this discussion to the Discussion section.

21.  Lines 408-429 and 435-463: The same as above… In the results section just please report your results derived based on your statistical approach.

22.  Line 535: It is better to become …difficult to be adopted

Discussion/conclusion:

23.  Lines 465-466, 1st sentence: I have the sense that the word “training” at the end of the sentence should be eliminated.

24.  Lines 517-519: Since you did not evaluate lean muscle mass and/or muscle hypertrophy particularly, in my point of view, you should not hypothesize that “there is little or no substantial hypertrophy”.

25.  Line 603: Please consider to replace the statement “must be” with something lighter.

26.  Your study, “aimed to explore the effectiveness of a 12-week circuit training in improving muscular endurance in primary school students”. However, at the same time you set criteria (i.e. BMI particularly) that made you to exclude n=123 kids. So, as far as I can understand, due to your exclusion criteria, you were not able to evaluate all the primary school students who wished to take part in your study BUT particularly those with BMI values above and below acceptable normal ranges. In addition, in your conclusion you claim that your proposed circuit training program is effective in improving local muscular endurance in school kids and also that “comprehensive school-based programs should increase health-related fitness components, such as strength …. However, almost 1/5 of the children were excluded from your study. Some boys perhaps were excluded due to their BMI values. Consequently, how you can generalize your results to be applicable in the curriculum of the physical education classes in primary schools?

27.  Considering also that primary school classes usually include both boys and girls. Could your results be generalized for girls as well? If your results can be applied/generalized only for boys, what do you propose for primary school girls when the boys will be performing your proposed circuit training program?

References:

28.  The references included in your study are in general not so recent. In my point of view, you should effectively update the list of your references with more recent articles and include also into your text updated information relevant to your subject study.

Round 2

Reviewer 2 Report

Dear authors,

Thanks a lot for addressing all my comments that have been raised during my initial review. I am happy enough with your responses and clarifications made to my comments and to the modifications accomplished within the text.

Some additional Minor comments:

11. Please check carefully once again the whole manuscript for minor spelling mistakes.

22. i.e. Line 764: Please replace the “noramal-weighted” with “normal-weighted”.